# *RB1* Sequence Variants in Retinoblastoma: Analysis of *RB1* Variants in a Database for Correlation with pRB Protein Domains and Clinical Presentation

**DOI:** 10.3390/biomedicines13112693

**Published:** 2025-11-02

**Authors:** Nicohol Tovar Martelo, Irene Szijan

**Affiliations:** Retinoblastoma, INIGEM, UBA-CONICET, Genetic Department, Pharmacy and Biochemistry School, Buenos Aires University, Buenos Aires 1113, Argentina

**Keywords:** ophthalmology, eyes, bilateral/unilateral retinoblastoma, RB1 variants, pRb domains

## Abstract

**Background**: Retinoblastoma (RB) is the most common pediatric ocular tumor that occurs due to the biallelic inactivation of the *RB1* tumor suppressor gene. RB may be unilateral or bilateral and is hereditary in 50% of cases. An inactivation of the *RB1* gene may occur due to gross rearrangements (20%) or due to small-length changes (80%): single nucleotide substitutions (SNVs) and insertions/deletions (INDELs). **Objectives**: Our objective was to study the frequency of the different RB1 variants present in patients with retinoblastoma and to correlate them with the functional domains of the pRb protein and with the clinical presentation. **Methods**: For this purpose, we analyzed all the clinically validated germline SNVs and INDELs annotated in the database. They were grouped into the pRb domains; contingency tables were made, and figures were constructed to compare the types of variants in the different domains between bilateral and unilateral patients. **Results**: The number of variants analyzed was 2103; 34% of them were nonsense, 34% INDELs, 22% splice-site and 10% missense. All these variants mainly gave rise to bilateral RB (88%); their frequency and distribution in relation to pRb domains varied between bilateral (Bi) and unilateral hereditary (Ug) RB. Nonsense variants occurred more frequently in Bi vs. Ug, whereas missense variants were more frequent in Ug vs. Bi. Indels and splice-site variants were not significantly different between Bi and Ug. The most frequent pRB location of variants was in the Pocket domain (the binding site of the E2F transcription factor). The slice-site of the consensus sequence most mutated was the first nucleotide of the donor, which is the driver of the splicing process. **Conclusions:** The highest percentage of variants in RB corresponded to nonsense substitutions and indels, mainly affecting the Pocket domain, which is the major functional site for the pRb regulatory process. These results indicate the predominance of the most pathogenic variants related to the bilateral presentation of retinoblastoma.

## 1. Introduction

Retinoblastoma (RB) is the most frequent ocular pediatric tumor. It occurs through the biallelic inactivation of the tumor suppressor *RB1* gene in one or more retinal precursor cells, inducing an uncontrolled cell division [1]. RB is a prototype of developmental tumors, since it occurs from prenatal age to 5 years old. It may be presented as unilateral (60%) or bilateral (40%), and rarely as trilateral (in both eyes and mostly in the pineal gland). RB has an approximate incidence of 1 in ~20,000 children born alive each year in the world [2]; the USA and European incidence was reported to be 12 and 4.0 per million, respectively [3,4]. The retinoblastoma tumor is hereditary in 50% of cases, including all bilateral cases and 15–25% of unilateral cases, with the majority of these latter cases being non-hereditary. In hereditary RB, the first *RB1* mutation is germline and the second is somatic; in non-hereditary RB, both mutations are somatic. Ten percent of hereditary RB is inherited and 30% arises “de novo”, with an average age at diagnosis of 1 year, while in the non-hereditary RB the average age at diagnosis is ~2 years or older [1].

The predisposition to RB is transmitted as an autosomal dominant trait with a penetrance of 90%, since the first mutation leads to the inactivation of both *RB1* copies and tumor development. Thus, individuals with a mutation have a 90% probability to develop retinoblastoma; accordingly, identification of the causative mutation is important to predict the risk for tumor development in patients’ relatives [5]. RB is potentially curable with an early diagnosis and has a risk of CNS and orbit/lymph-node metastasis in late cases; thus, early diagnosis is critical for survival and eye preservation in children who carry an *RB1* mutation. The prognosis of patients with RB is good in developed countries, with a survival rate of almost 100%; however, most children in the world do not have a good prognosis, and a large percentage die owing to spread of the disease because of a late diagnosis. The survival rate is mainly influenced by socioeconomic and cultural factors. In Latin America, the prognosis improved up to 80% thanks to early diagnosis and effective treatments; in Argentina, RB presents a survival rate of 90%. The presumptive diagnosis of RB signs such as leukocoria or strabismus are early enough to save lives.

The human *RB1* gene was the first tumor suppressor gene isolated, and it is expressed in many tissues [6]. The protein encoded by this gene, pRb, contains several functional domains, including the highly conserved Pocket domain that interacts with E2F transcription factors, preventing the expression of genes for G1 to S transition. Mutations in the *RB1* gene disrupt the structure and function of pRb, leading to the deregulation of cell proliferation. *RB1*-inactivating aberrations may be (1) small sequence variants (80%), including single nucleotide substitutions and deletion/insertions (indels), or (2) gross rearrangements (20%). Most of them are null mutations, leading to an absence of the pRb protein; they account for 90% of all *RB1* mutations and include nonsense, frameshift and splice-site variants. On the other hand, missense, in-frame and promoter variants are infrequent. Gross rearrangements occur by deletions or duplications of part of *RB1* (several exons) or of the whole *RB1* gene.

This study is a continuation of our search for causative mutations in RB patients. Our objective was to study the frequency of the different RB1 variants present in patients with retinoblastoma and correlate them with the functional domains of the pRb protein and with the clinical presentation. For this purpose, we analyzed the clinically validated germline SNVs and INDELs annotated in the database http://rb1-lovd.d-lohmann.de (Accessed on 15 November 2021). The frequency of different variants and their distribution in pRb domains varied between bilateral and unilateral patients. Our results indicate the predominance of the most pathogenic variants related to the bilateral presentation of retinoblastoma.

## 2. Materials and Methods

An analysis of the small size variants along the *RB1* gene, displayed in the http://rb1-lovd.d-lohmann.de database, was carried out. Since most of the clinically validated variants in this database are of germline origin (73%), we analyzed these types of SNV and INDEL variants that occur in bilateral and unilateral hereditary RB patients. In addition, some data obtained in our laboratory (50, 2% of total), using Sanger sequencing, whole exome sequencing (WES) and MLPA, were included. The study methodology consisted of enumerating the different classes of variants in the exons of each functional region of the pRb protein in bilateral and unilateral hereditary patients and creating tables for each of the classes. The percentages were then calculated, and the figures were created using the Prisma program.

Informed consent for genetic analysis was signed by parents of the affected children according to the principles of the Declaration of Helsinki. The study was approved by the Ethics Committee of “Hospital de Clinicas Jose de San Martin” Universidad de Buenos Aires, Buenos Aires. Argentina, 28 November 2005 (code number: 102-05).

The detailed methodology was as follows. Nonsense, missense, indel and splice-site variants were enumerated in all exons of the RB1 gene in the database and grouped into the pRb domains (RbNA, promoter-exon 8; Spacer, exon 8; RbNB, exons 8–11; RbIDL, exons 11–12; Pocket A, exons 12–18; RbPL, exons 18–19; Pocket B, exons 19–23; RbC, exons 23–27). Next, the different types of nonsense variants (CGA > TGA, CAG/CAA > TAG/TAA and other non-C > T substitutions) were determined in the Pocket A domain and compared with the average of these variants in the other pRb domains in all bilateral and unilateral patients. In the next place, the total number of nonsense, missense, indel (frameshift) and splice-site variants (donor, sites 1–5, acceptor, sites 1–3) in the pRb domains were compared between bilateral and unilateral patients. For this purpose, contingency tables were created with the results of the variants in the different domains, combined or individual, according to the type of variant, in bilateral and unilateral patients. Based on these tables, figures were constructed using Prisma software (Prism 6, version 6.01).

Statistical assay: The program GraphPad Prism 6, version 6.01 (GraphPad Prism Software, La Jolla, CA, USA) was used. Contingency tables were made using Fisher’s exact test and Chi square test (with a Yates correction when corresponding). The latter test was used for an approximate and not an exact calculation. Those with a *p* < 0.05 were considered statistically significant. The statistical hypotheses for each case were based on the variables studied; for example, the type of variant is associated with its location in pRb domains and the tumor laterality.

**Null hypothesis** **(H0).**
*The variants are not associated (variant´s independence).*


**Alternative hypothesis** **(H1).**
*The variants are associated (variant´s dependence).*


## 3. Results

The number of variants analyzed was 2103. The highest percentage of variants corresponded to the nonsense and indel classes, 34%, followed by splice-site variants, 22%, and the lowest percentage was for the missense variants, 10%. Results from bilateral patients were predominant in the database, reaching 94% of nonsense variants, 90% of indels, 85% of splice-site variants and 71% of missense. Therefore, to compare the frequency of different variants between bilateral and unilateral hereditary retinoblastoma, percentages of each type of variant relative to the total were used. The frequency of different variants and their distribution in pRb domains may vary between bilateral and unilateral patients.

Nonsense variants in bilateral plus unilateral hereditary cases were divided into the following classes to study their distribution in pRb domains: (i) CGA > TGA, (ii) CAA/CAG > TAA/TAG and (iii) other substitutions, non C > T. The occurrence of CGA > TGA substitutions was by far the highest compared with the other substitutions (76%), and their presence in the Pocket A domain was significantly higher (88%) compared with the average value in other pRB domains (12%) (Figure 1). To compare the presence of nonsense variants in pRB domains between bilateral and unilateral patients, they were analyzed in the following combined pRb domains: (i) the RbN region and (ii) the RbC region, including the Pocket domain. The distribution of nonsense variants between RbN and RbC pRb domains was similar in bilateral and unilateral patients; most of them (60%) occurred in the Pocket domains (Figure 2).

Missense variants occurred with a significantly higher frequency in Pocket B than in other domains in bilateral patients (58%), while in unilateral patients these variants were significantly higher in RbN (48%) compared to other domains (*p* value: 0.00002) (Figure 3). A total of 10% of missense mutations also caused an alteration in the splicing mechanism in bilateral and unilateral patients. This alteration was due to the location of the variant in the last nucleotide (64%), or in one of the last nucleotides (18%) of an exon, or to the creation of a consensus splicing motif (GT) inside the exon 7 (18%).

The frequency of indel variants in bilateral patients was similar to that of nonsense variants (34%), while in unilateral patients the indel frequency was higher than that of nonsense (29% vs. 17%); however, the difference was not significant. By far, the vast majority of indels were frameshift: 99% in bilateral patients and 85% in unilateral patients. The percentage of deletions was higher than that of insertions/duplications in both bilateral and unilateral RB (66% vs. 30%, combined del and ins 4%). The distribution of indels between pRb domains showed a higher incidence in RbN and Pocket A than in other domains in bilateral patients, while in unilateral patients indels occur more frequently in RbN than in other domains (Figure 4).

Splice-site variants corresponded mainly to the Pocket domain region in the pRb protein in both bilateral and unilateral hereditary patients. The analysis of each Pocket domain separately showed a significantly higher frequency of variants in Pocket A with respect to other domains in unilateral patients, and it was significantly greater than in bilateral patients. In the latter group, the variants were distributed similarly between the RbN domain, Pocket A and Pocket B. (Figure 5). The site of the consensus sequence most mutated was the first nucleotide of the donor (52% of total donor sites), while the first nucleotide of the acceptor was less mutated (30% of the total acceptor sites) in the sum of bilateral and unilateral patients. Moreover, the frequency of variants in the donor site vs. the acceptor site was 75% vs. 25% in both bilateral and unilateral patients.

A comparison of the different classes of variants between bilateral and unilateral hereditary patients showed a significant difference between the two groups. Nonsense variants occurred with a significantly higher frequency in bilateral than in unilateral patients, while missense variants were significantly more frequent in unilateral than in bilateral patients. On the other hand, the indel variants were more frequent in bilateral patients and the splice-site variants were more frequent in unilateral patients, but the differences were not significant (Figure 6).

## 4. Discussion

Most of the variants present in the database correspond to germline mutations in bilateral patients. This is explainable given that germline mutations, detectable in blood, occur in all bilateral and only in 15 to 25% of unilateral patients. In these last patients, the predominant somatic mutations are detectable in the tumor (not frequently available).

The most frequent changes in the *RB1* gene sequence were single nucleotide substitutions and indels; the former included nonsense, missense and splice-site variants. A homologation of sequence variation in the *RB1* gene to changes in the pRb protein may indicate the effect of *RB1* variants on pRb function. This data may clarify the impact of variants on retinoblastoma presentation.

The pRb protein consists of several domains that serve different functions. The RbN A and RbN B domains, in the N-terminal region, and the Pocket A and Pocket B domains, in the central and c-terminal regions, are structured domains that play a role in the binding of transcription factors such as the E2F family. The Spacer, RbIDL, RbPL and RbC domains are disordered, located between the structured domains, and contain phosphorylation sites. The phosphorylation process by cyclin-dependent kinases causes a conformational modification in pRb, which changes the spatial relationship between domains and induces the release of E2F transcription factors [7]. These factors are involved in the control of G1/S transition, mediated by the transcriptional activation of the genes required for cell cycle progression and DNA replication. The interaction between pRb and E2F transcription factors may also be inhibited due to the action of several viruses or due to variations in the *RB1* sequence.

Nonsense variants, the most frequent substitutions, include base changes in the codons of arginine, glutamine and other amino acids. The most common nonsense variants are C > T substitutions in CpG dinucleotide sequences, which constitute 32% of all variants [8]. C > T substitution is produced mainly by oxidative deamination, mediated by cytosine methylation at the CpG dinucleotide, which is a very common epigenetic mechanism and the most common post-synthetic change in the CpG dinucleotide. The substitution of cytosine for thymine may occur through another mechanism, a transient misalignment of the DNA strand at the point of replication, given that DNA is a dynamic structure and can adopt a variety of conformations; this is common to all nucleotide substitutions, nonsense, missense and splice-site.

The generation of indels is also due to the misalignment of the two DNA strands by a slippage mechanism, but in a different way. While base substitution occurs due to a temporary misalignment, with subsequent realignment, indels originate from a permanent misalignment of the DNA strands. The result of this difference is the change in a single base in the case of nucleotide substitution and the loss or gain of one or more bases, which can lead to frequent reading frame shifting in the case of indels [9].

Splice-site variants occurred due to the base substitution mechanism in the consensus splice sequences that include the canonical GT (donor) and AG (acceptor) sites plus flanking nucleotides, encompassing up to approximately five bases [10,11]. By far, the most frequent site of variation in retinoblastoma was the first nucleotide of the donor site, which is the driver of the splicing process [12]. This indicates that the predominant splicing variants are the most pathogenic.

The main pRb location sites of nonsense, missense, indel and splice-site variants were the functional domains RbN and Pocket (8–48%), leading to a loss of pRb function, while the location of variants in disordered regions, among structured domains, was lower (2–10%) in both unilateral and bilateral patients. pRb protein binds E2F transcription factors in the cleft between Pocket A and Pocket B, which maintains contact with the RbN domain. The pathogenicity of variants located in these domains is due to a change in pRb conformation, caused by sequence variation, that prevents E2F binding.

Missense mutations in the Pocket domain are relevant because mRNA with a missense mutation is not degraded by “Nonsense mediated decay” [13]; thus, it is translated, generating a pRb protein with an altered structure at the E2F binding site, leading to the release of E2F from the pRb-E2F complex [7]. Missense substitutions in sequences involved in the splicing mechanism reinforce the alteration in the pRb structure, produced by a change in amino acids, with that of splicing disturbance.

The similar distribution of nonsense variants between pRb domains in bilateral and unilateral patients suggests that nonsense variants function similarly in bilateral and unilateral hereditary patients. On the other hand, the distribution of missense and indel variants in pRb domains was different between bilateral and unilateral retinoblastoma. The highest frequency of these variants occurred in Pocket domains in bilateral patients and in RbN domains in unilateral patients. This difference indicates that alterations in the domain critical for pRb function predominate in bilateral patients compared to alterations in structured but less critical regions in unilateral patients. In-frame indels, which are benign variants, occur with a higher incidence in unilateral patients (15%) than in bilateral ones (1%). The pRb domains most affected by splice-site variants were Pocket A and B in both bilateral and unilateral patients.

The prevalent location of *RB1* variants in the Pocket domain correlates with the high pathogenicity of variants occurring in retinoblastoma. The difference in severity between bilateral and unilateral retinoblastoma agrees with the different pRb domains where the variants occurred, namely, mostly in the Pocket domain in bilateral patients and in the RbN domain in unilateral patients. Other molecular changes, such as in epigenetic and regulatory mechanisms [14,15], may contribute to the phenotypic differences between unilateral and bilateral retinoblastoma.

A comparison of variant types between bilateral and unilateral patients showed that the most frequent were nonsense in bilateral patients and missense in unilateral patients. These results indicate that, in bilateral patients, the variants that have the highest impact on retinoblastoma development prevailed, since they cause an absence of the pRb protein. On the other hand, in unilateral patients, the most frequent were the variants that originate changes in the pRb sequence, decreasing only its function.

The results obtained are an approach to the study of genetic variants in patients with retinoblastoma, since the database analyzed is not a complete list of variants that occurred in the *RB1* gene. There are other databases and there are also unregistered variants; therefore, the data analyzed can be used as a sample in the total population of RB cases.

The results obtained could contribute to a better understanding of retinoblastoma and could be useful for genetic counseling. The difference in the type of genetic variants, correlated with the appropriate functional domains of pRb, between bilateral and unilateral patients is important for the prognosis of unilateral patients, since bilateralization may occur in the latter. It is also worth noting that the more benign variants present in less functionally significant domains of pRb, such as changes in the promoter sequences or missense variants, both corresponding to the RbN domain, are less penetrant. This means that carriers of these variants may have a less severe clinical presentation (unilateral form) or not develop tumors at all. These concepts are important to consider in genetic counseling.

## Figures and Tables

**Figure 1 biomedicines-13-02693-f001:**
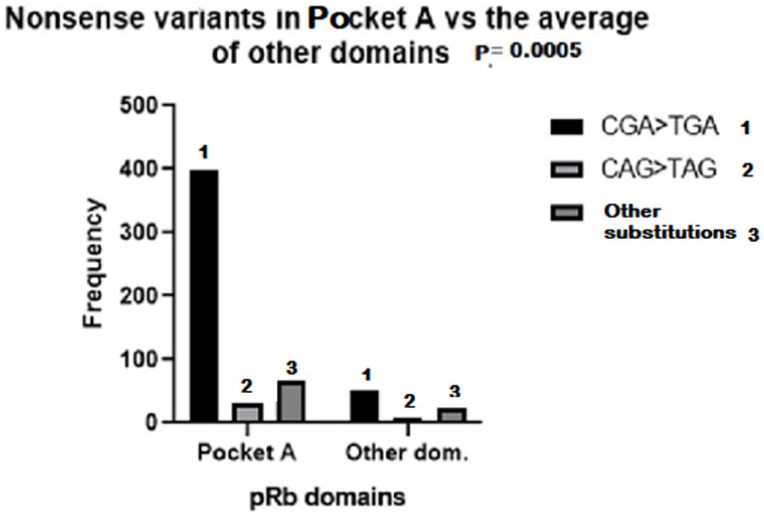
Frequency of CGA > TGA (1), CAG > TAG (2) and other nonsense substitutions (3) in RB1 regions corresponding to pRB Pocket A and other domains, in which the frequency was calculated as the average of the sum of all domains except Pocket A. (Variants in Pocket A domain vs. the average of other domains).

**Figure 2 biomedicines-13-02693-f002:**
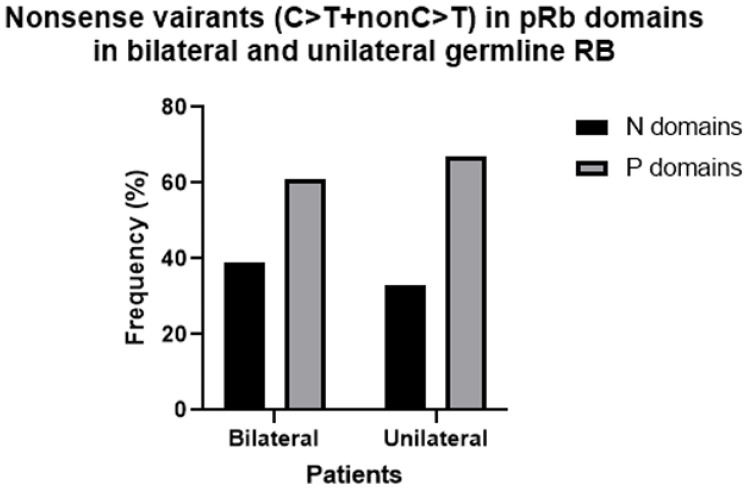
Frequency of all nonsense variants in Pocket (P) and RbN (N) domains in bilateral vs. unilateral hereditary retinoblastoma.

**Figure 3 biomedicines-13-02693-f003:**
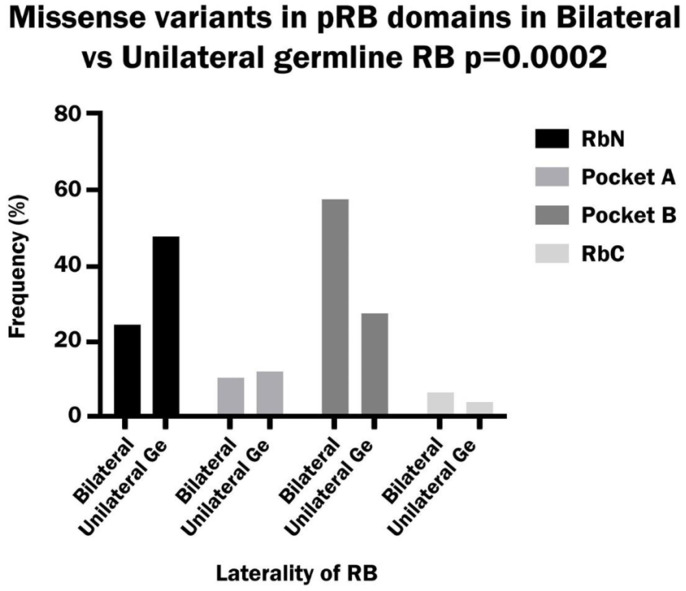
Frequency of missense variants in the RbN, Pocket A, Pocket B and RbC domains of pRB in bilateral vs. unilateral hereditary retinoblastoma.

**Figure 4 biomedicines-13-02693-f004:**
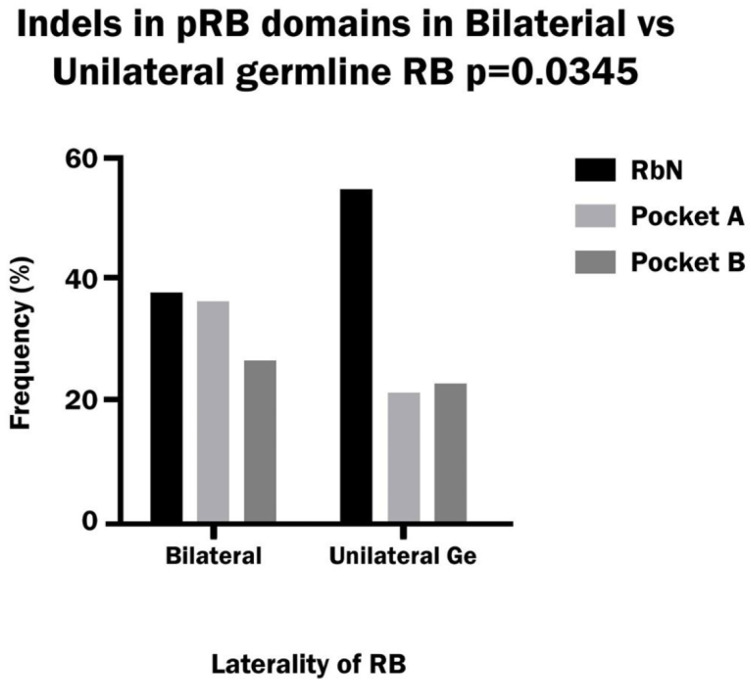
Frequency of Indels variants in the RbN, Pocket A and Pocket B domains of pRB in bilateral and hereditary unilateral retinoblastoma.

**Figure 5 biomedicines-13-02693-f005:**
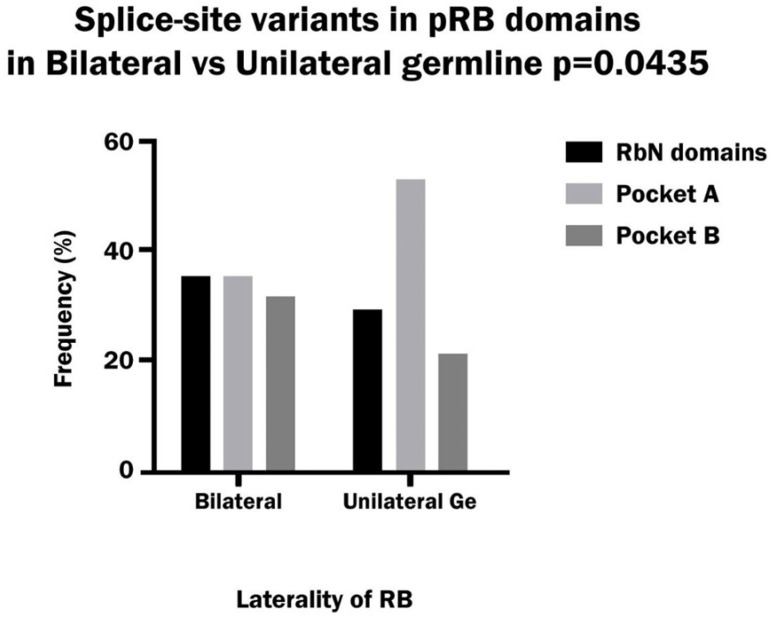
Frequency of splice-site variants in the RbN, Pocket A and Pocket B domains of pRB in bilateral and hereditary unilateral retinoblastoma.

**Figure 6 biomedicines-13-02693-f006:**
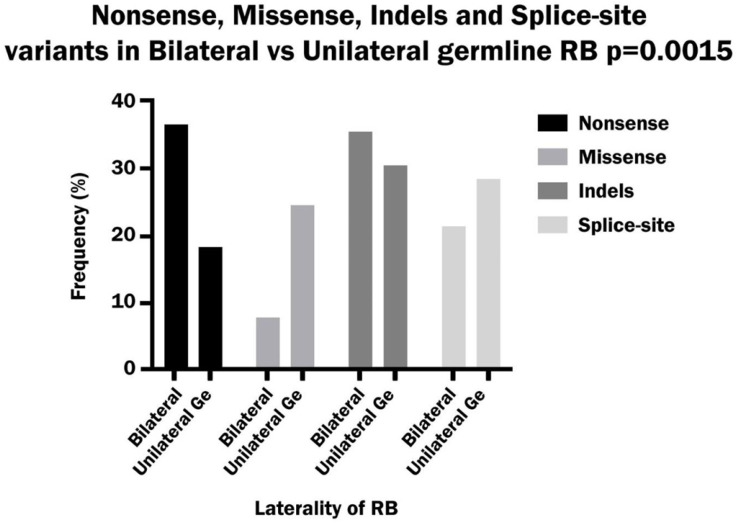
Comparison of the frequency of nonsense, missense, Indels and splice-site variants in bilateral and hereditary unilateral retinoblastoma.

## Data Availability

Data supporting the reported results can be found in http://rb1-lovd.d-lohmann.de (Accessed on 15 November 2021).

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
