# Peer review of "RB1 Sequence Variants in Retinoblastoma: Analysis of RB1 Variants in a Database for Correlation with pRB Protein Domains and Clinical Presentation"

_biomedicines, 2025, doi:10.3390/biomedicines13112693_

Round 1
Reviewer 1 Report
Comments and Suggestions for Authors
The manuscript by Tovar-Martelo Nicohol et al. addresses an important and clinically relevant topic: the distribution of RB1 sequence variants and their correlation with functional pRB protein domains and clinical presentation in retinoblastoma. The study is based on a large dataset (2,103 variants) and complements it with the Authors’ own sequencing data.
Recommendations for Authors:
1. Lack of a clearly stated hypothesis in the Introduction; the rationale behind the analysis could be better framed.
2. Clinical significance is underdeveloped: the discussion does not fully explain how the findings could improve diagnostics, genetic counseling, or prognosis.
3. Data presentation: figures are of poor quality, lack sample size annotations (N) for better comparison.
4. Methods section: The selection criteria for variants from LOVD are not sufficiently detailed. Were only clinically validated entries analyzed? The contribution of the authors’ own data is unclear (how many variants, proportion to database).
5. Language and style: the text contains grammatical and stylistic errors that require correction.
6. References: some inconsistencies and typos in author names and formatting.
Comments on the Quality of English LanguageThe text contains grammatical and stylistic errors that require correction.
Author Response
- Lack of a clearly stated hypothesis in the Introduction; the rationale behind the analysis could be better framedReplyReply:
- Clinical significance is underdeveloped: the discussion does not fully explain how the findings could improve diagnostics, genetic counseling, or prognosis
- Data presentation: figures are of poor quality, lack sample size annotations (N) for better comparison.
- Methods section: The selection criteria for variants from LOVD are not sufficiently detailed. Were only clinically validated entries analyzed? The contribution of the authors’ own data is unclear (how many variants, proportion to database).
- Language and style: the text contains grammatical and stylistic errors that require correction.
- References: some inconsistencies and typos in author names and formatting.
Comments on the Quality of English Language
The text contains grammatical and stylistic errors that require correction.
Reply:: Introduction: The objective was added in the final paragraph
Materials and Methods: The criteria for selecting variants in LOVD were detailed and the number of results obtained by the authors
Results: The quality of the figures has been improved by indicating the meaning of the different bars
Discussion: Clinical significance is indicated, explaining how the findings could improve genetic counseling and prognosis
Language and grammatical errors were revised
Reviewer 2 Report
Comments and Suggestions for Authors
Dear authors,
Firstly, I would like to congratulate you on your work.
I have a few suggestions that I hope will improve the academic value of your work and make your research easier to read in the future.
In the abstract, I suggest stating the objective and study design, which are clear in the manuscript but not in the abstract. Understanding your objective is necessary to consider whether your conclusion is appropriate to the context of your study. Understanding your study design will help us analyze the level of evidence that your research can provide.
As keywords, I suggest adding: ophthalmology; eyes
The introduction is adequate and the references are correct. However, in the last paragraph, I suggest explicitly stating your objective: what you have done, developing it directly in methods. For example, stating that you have analyzed a particular database. All of this in methods.
But first, please clearly state YOUR objective.
***Improve and provide more detail in the methods section.
The results are well presented, but in the version I evaluated, the figures were not very clear. Consider this for improvement in your final version.
Discussion
It is well presented in the context of the interpretation of your results. However, at the end, I miss one or two more specific sentences by way of conclusion. I suggest adding or making this explicit.
Finally, dear authors, the work you have done is relevant and makes an interesting contribution to the scientific community, so I believe it should be published and made available to our peers. But to do so, I suggest that some methodological aspects be made explicit, both in the abstract and in the manuscript, to facilitate reading not only by experts in the field, but also by those who need this material to continue improving the study and understanding of this pathology.
Congratulations.
Author Response
1. In the abstract, I suggest stating the objective and study design, which are clear in the manuscript but not in the abstract. Understanding your objective is necessary to consider whether your conclusion is appropriate to the context of your study. Understanding your study design will help us analyze the level of evidence that your research can provide
The changes in the text were in italics to find them easily.
:The objective and study design as well as methodological details were added.
2. As keywords, I suggest adding: ophthalmology; eyes
Suggested keywords have been added
3..Improve and provide more details in the method section
: Methodological aspects were corrected to make them more explicit
4. The results are well presented but the figures were not very clear. Considere this to improve your final versio
The quality of the figures has been improved by indicating the meaning of the different bars
5.Discussion It is well presented in the context of the interpretation of your results. However, at the end, I miss one or two more specific sentences by way of conclusion. I suggest adding or making this explicit.
Clinical significance is indicated, explaining how the findings could improve genetic counseling and prognosis The changes in the text were in italics to find them easily
Reviewer 3 Report
Comments and Suggestions for Authors
The paper analyzes variants in an existing databases and correlates their impact and position with the functional domain of the RB1 retinoblastoma protein.
While the paper can be of interest and a good base for future studies, it lacks depth and quality of presentation.
For the record, the journal or editor should request line numbering to help the reviewing process.
There are several formatting errors like lack of punctuation in the abstract or different fonts in the main body of text. The overall paper feels rushed.
The introduction lacks of citations for the first 4-5 lines. Many statements made with no references at all. This happens again mid-introduction. The in the end. See
"The human RB1 gene was the first tumor suppressor gene isolated, and it is expressedin many tissues." No references whatsoever.
Please cite or rewrite it entirely.
The authors keep calling "genetic variants" "mutations". I invite the authors to tell apart the two. The mutation is the event, a protein can be mutated, an individual a mutant, a given allele within a population is a genetic variant.
In results, the type-of-variant discourse needs a table. The figures of the graphs are all over the place in size and resolution, to an unacceptable level.
There is no effort in classifying the missense variants. That alone would require a whole table. More importantly, a serious statistic must be shown to indicate the correlation between a variant type and phenotypes.
The discussion loses its focus quite early and starts to digress at the "Generation of indels is also due to misalignment of the two DNA strands by slippage...." - there is no need to explain the whole mechanism, this feels like padding.
The idea behind the paper can be interesting but the execution is SEVERELY LACKING on all standpoints.
Author Response
- There are several formatting errors like lack of punctuation in the abstract or different fonts in the main body of text. The overall paper feels rushed
- The introduction lacks of citations for the first 4-5 lines. Many statements made with no references at all. This happens again mid-introduction. The in the end. See
"The human RB1 gene was the first tumor suppressor gene isolated, and it is expressedin many tissues." No references whatsoever.
Please cite or rewrite it entirely. - The authors keep calling "genetic variants" "mutations". I invite the authors to tell apart the two. The mutation is the event, a protein can be mutated, an individual a mutant, a given allele within a population is a genetic variant.
- In results, the type-of-variant discourse needs a table. The figures of the graphs are all over the place in size and resolution, to an unacceptable level.
- There is no effort in classifying the missense variants. That alone would require a whole table. More importantly, a serious statistic must be shown to indicate the correlation between a variant type and phenotypes
- The discussion loses its focus quite early and starts to digress at the "Generation of indels is also due to misalignment of the two DNA strands by slippage...." - there is no need to explain the whole mechanism, this feels like padding.
- .Reply: 1. Formating errors were revised.
- 2. Introduction Several citations were added
- 3. A mutation is a base change in DNA. A genetic variant is also a base change, therefore either of the two can be used to name a change of one base or several bases
- 4.. I don't think it's necessary to include tables in the manuscript since the figures are more demonstrative of the results.
- 5. I don't think it's necessary to include table. I don´t think it is necessary other statistic mehtods as the ones used are adequate for an approche to study genetic variants using those annoted in one data base.
- 6. The explanation of the possible mechanism of changes in DNA is useful to understand how easily alterations can occur.
Round 2
Reviewer 1 Report
Comments and Suggestions for Authors
The article has been corrected with previous comments.